Boolean networks; computer modelling; plant networks; stomatal closure.

**Author for correspondence:**
W.-J. Rappel,
E-mail: rappel@physics.ucsd.edu

# Boolean modelling in plant biology

Aravind Karanam and Wouter-Jan Rappel

Department of Physics, University of California, San Diego, La Jolla, California 92093, USA

## Abstract

Signalling and genetic networks underlie most biological processes and are often complex, containing many highly connected components. Modelling these networks can provide insight into mechanisms but is challenging given that rate parameters are often not well defined. Boolean modelling, in which components can only take on a binary value with connections encoded by logic equations, is able to circumvent some of these challenges, and has emerged as a viable tool to probe these complex networks. In this review, we will give an overview of Boolean modelling, with a specific emphasis on its use in plant biology. We review how Boolean modelling can be used to describe biological networks and then discuss examples of its applications in plant genetics and plant signalling.

## 1. Introduction

Many cellular processes in biology are controlled by a large number of components that are part of complex signalling networks (Kitano, 2002). Examples include the pathways controlling cell polarity, cell motility, cell division and differentiation as well as the gene networks that underlie a myriad of biological processes. The biological function in question frequently arises out of the connections and dependencies among physical and chemical processes which may be relatively simple and well understood. Technological advances in the last few decades have contributed to a proliferation of data at the level of individual genes and metabolites (International Human Genome Sequencing Consortium, 2001; Tyers & Mann, 2003), paving the way for synthesising the knowledge to produce a systems-level understanding.

Models of biological networks attempt to recast the systems in a mathematical form and their level of detail depends on the amount of available data as well as its requirements (Karlebach & Shamir, 2008). In its optimal form, quantitative modelling can replace often laborious experiments by carrying out in silico experiments during which one or more components of the pathway or the interactions between components are altered. Even if this is not possible, modelling can often reveal the role of a particular component in the pathway and can, thus, predict the effect of removing or making it constitutively active.

Constructing models for biological pathways requires knowledge about their topology. In other words, one needs to know whether component *A* affects component *B*. This is equivalent to answering the question whether *B* is downstream or upstream of *A*. Furthermore, the 'sign' of the interaction between these components is required: does *A* activate (corresponding to a positive interaction) or inhibit (negative interaction) *B*? Ideally, one would also like to know the strength of the interaction: how much will *B* increase or decrease when *A* is present?

For pathways in which all connections and strengths are known, it is possible to construct a mathematical model that represents concentrations of the pathway components as continuous quantities that can take on all positive values. This type of model can provide significant insights, particularly for small systems made up of a handful of simple reactions where all the interactions are known (Ouellet et al., 1952; Pollard, 1986). For these systems, parameters like rate constants, dissociation and association constants can be inferred by monitoring the formation of the product or the decay of the substrate. Often, however, and especially for pathways that contain many components, it is not possible to quantify the type of interaction and the strength between the different components. After all, quantifying this for, say, *A* and *B* typically requires a systematic variation of the level *A* and measuring the response in *B*. This type of experiment is not always possible for all components and models with a large number of unknown parameters, and interactions can quickly lose their predictive and mechanistic value.

An alternative to creating continuous models is to construct Boolean models (Schwab et al., 2020). In a Boolean model, each element (alternatively called a node) can only take on one of two values: 0 and 1. The dynamics of these nodes are no longer determined by solving equations that involve rate constants but are updated using logical operations. These operations encode the connections between the different components using the elementary logical functions: identity, AND, OR and NOT. A Boolean network is then obtained by connecting a number of such nodes in a meaningful manner. Despite these significant simplifications, Boolean networks have been shown to be able to provide insights into genetic networks (Herrmann et al., 2012; Kauffman, 1969b; Shmulevich et al., 2002b; Thomas, 1973), protein networks (Bornholdt, 2008) and cellular regulatory networks (Lau et al., 2007; Li et al., 2004). More importantly, from a practical point of view, Boolean models have several advantages: they can be simulated relatively quickly, even on daily-use desktop computers, and several software packages are freely available (Karanam et al., 2021; Schwab et al., 2020); modifying the network and simulating variants of the original network are easy tasks, and thus Boolean modelling can be used to (a) generate hypotheses that can be tested by experiments and (b) systematically explore variants of a network that 'predict' or lead to an observed phenotype. These ideas have been explored in several studies as will be described later (Karanam et al., 2021; Maheshwari et al., 2019).

In this review, we focus on Boolean modelling in plant biology. We start with a brief overview of Boolean logic and how one can deduce a Boolean network from rate equations as well as from experimental data. We then discuss software packages that can be used to simulate Boolean networks, after which we discuss applications of Boolean modelling to gene regulatory networks (GRNs) in plants. We then review how Boolean modelling can be used to probe the pathways in guard cells that lead to stomatal closure in response to the plant hormone abscisic acid (ABA) and carbon dioxide ($CO_2$), and end with a brief conclusion and outlook.

## 2. Boolean logic and networks

In this section, we will first describe in more detail how Boolean equations are evaluated, provide a simple example, and show how truth tables are a convenient way to analyse and comprehend small Boolean networks. We will then describe how a Boolean network can be constructed from experimental data and describe the various updating schemes developed for this type of network. We will also show how one can translate rate equations into Boolean equations and finish by discussing available software for the simulation of Boolean networks.

### 2.1. Truth tables

As we mentioned in Section 1, the nodes in a Boolean network can only take on values 0 (OFF) and 1 (ON). The ON state of a variable corresponds to high activity or concentration and the OFF state corresponds to low activity or concentration. The interactions between the nodes are given by a combination of the logical functions AND, OR and NOT acting on the input nodes that feed into the output node. The future state of the output node (say at time $t+1$) is obtained by evaluating its corresponding Boolean function that takes the current states (say at time $t$) of its input nodes as inputs. To simplify notation, we write the output node and the update rule together as an equation, commonly known as the update equation of the output node. We do not explicitly specify time because (a)

the update rules do not change with time and (b) the states of the input nodes specify, through the update equation, the state of the output node in the succeeding time step only.

As a simple example, consider the activation of gene $B$ by a transcription factor $A$. In this case, when the concentration of $A$ is high, the gene is ON, whereas when it is low, $B$ is OFF. This process can be mathematically expressed using an ordinary differential equation, which describes the rate of change of $B$, $dB/dt$, as a function of the concentration of $A$. In its simplest form, this differential equation is written as

$$\frac{dB}{dt} = f(A) - \gamma B.$$

Here, $\gamma$ is a degradation constant, determining how $B$ is removed, and the function $f(A)$ describes how the production rate of the gene depends on the transcription factor concentration $A$. This function is often taken to be a Hill function $f(A) = \beta A^n/(A^n + K^n)$, with $n$ the (integer) Hill coefficient, $\beta$ the maximum production rate and $K$ the activation coefficient. If we take $n$ to be very large, we can approximate $f(A)$ to be a so-called step function: $f(A) = 0$ if $A < K$ and $f(A) = \beta$ if $A \geq K$. Thus, when $A < K$, $B$ will be 0, while for $A > K$, the time dependence of $B$ is found by solving the differential equation

$$\frac{dB}{dt} = \beta - \gamma B. \tag{1}$$

The steady-state value, achieved after a long time, can be found by setting the left-hand side of this equation to zero, resulting in $B = \gamma/\beta$. Furthermore, assuming that $A$ is set above the threshold value $K$ at $t = 0$, the solution of this equation can be found to be $B(t) = \frac{\gamma}{\beta}(1 - e^{-\gamma t})$. This solution is shown in Figure 1a, where we plot $B$ as function of time for different values of the degradation constant and using $\gamma/\beta = 1$ for simplicity. When the transcription factor is turned ON, $B$ approaches its steady-state value at a timescale that depends on $\gamma$. In this description of gene activation, $B$ can take on all possible values between 0 and 1.

Consider, on the other hand, a simplification of the model in which $A$ and $B$ can only take on values of 0 or 1 and in which the presence of $A$ causes an instantaneous rise in $B$ from 0 to 1. This model can be simply formulated without any parameters by a Boolean equation, which defines how the value of $B$ is updated given the value of $A$. This equation can be compactly written as

$$B^* = A, \tag{2}$$

where we have adopted the convention that the variable with an asterisk is being updated. In other words, if $A = 0$, then $B$ is updated to 0, independent of its current state. If $A = 1$, on the other hand, $B$ is updated to 1, again independent of its current state. The time course of this Boolean equation is shown in red in Figure 1a, where $A$ is changed from 0 to 1 at $t = 0$. In contrast to the differential equation, $B$ is immediately turned ON when $A$ is set to 1.

The above example is very simple and does not involve any of the elementary Boolean functions. To illustrate these functions, let us now consider the nodes $A$ and $B$ as the input nodes and $C$ as the output node. A useful way to characterise the logical operations is to construct the so-called truth tables, which list the output values for all possible combinations of input values. The truth tables for the elementary logical functions are listed in Figure 1b,c. For example, the AND function ($C^* = A$ AND $B$) only returns $C = 1$ if both $A$ and $B$ are ON and will return 0 for all other input combinations. Similarly, an OR gate returns an output of 1 if at least one or both of the inputs is 1. Obviously, the identity gate copies the current state of the only

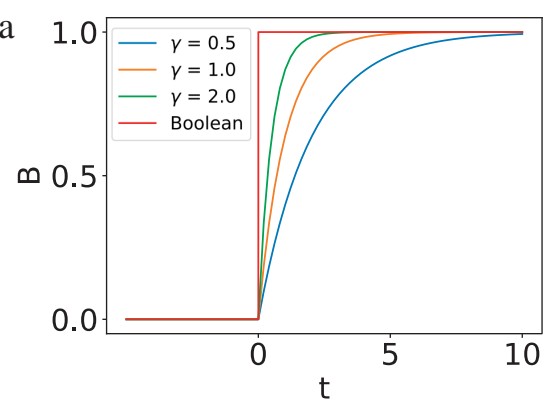

a

b

| A | C*= A | C*= NOT A |
|---|---|---|
| 1 | 1 | 0 |
| 0 | 0 | 1 |

c

| A | B | C*= A OR B | C*= A AND B |
|---|---|---|---|
| 0 | 0 | 0 | 0 |
| 0 | 1 | 1 | 0 |
| 1 | 0 | 1 | 0 |
| 1 | 1 | 1 | 1 |

d

| A | B | C | NOT A | B OR C | X*= (NOT A) AND (B OR C) |
|---|---|---|---|---|---|
| 0 | 0 | 0 | 1 | 0 | 0 |
| 0 | 0 | 1 | 1 | 1 | 1 |
| 0 | 1 | 0 | 1 | 1 | 1 |
| 0 | 1 | 1 | 1 | 1 | 1 |
| 1 | 0 | 0 | 0 | 0 | 0 |
| 1 | 0 | 1 | 0 | 1 | 0 |
| 1 | 1 | 0 | 0 | 1 | 0 |
| 1 | 1 | 1 | 0 | 1 | 0 |

**Fig. 1.** (a) Comparison of the output of a continuous model [equation (1)] and a Boolean model [equation (2)] for the activation of a gene. In the former, the output can take on any value between 0 and 1 and depends on the model parameters, whereas in the latter, the output is either 0 or 1 and is independent of parameters. (b–d) Truth tables of elementary Boolean functions. (b) Identity gate, which copies the value of the input to the output; NOT gate, which copies the inverted value of the input to the output. (c) OR and AND gates, which take two inputs. (d) An example of a Boolean function that is a combination of the elementary functions. The output $X$ can be determined by evaluating the parts recursively.

input node to the future state of the output node. Similarly, the NOT gate only has a single node as input and it inverts the current state of the node. It can be shown that by compounding these three elementary gates it is possible to encode all Boolean functions (Mano & Kime, 1997), including some commonly encountered ones in electronics, such as XOR (exclusive OR). In Figure 1d, we show the truth table of the compound Boolean function $X^* = (\text{NOT } A) \text{ AND } (B \text{ OR } C)$. This table also illustrates how the output node $X$ is updated by evaluating parts of the function recursively.

### 2.2. Update rules

Once a Boolean network is constructed, the nodes are updated following a particular update scheme. This is a choice the investigator needs to make because a Boolean model contains neither a natural timescale nor a specified order in which the reactions of the model take place. In the existing literature on Boolean models, three types of update schemes have been used: synchronous, asynchronous and probabilistic update schemes (Schwab et al., 2020). In synchronous Boolean models, all the components are updated at the same time, that is, the states of all the nodes at time step $t + 1$ are determined by their states at time step $t$ (Espinosa-Soto et al., 2004; Fauré et al., 2006; Garg et al., 2008; Remy et al., 2006). This also means that the evolution of a synchronous Boolean model is deterministic: a particular input will always result in the same output.

An asynchronous Boolean model orders the updates of the nodes one after another in either a pre-determined or a stochastic manner. There are a number of ways to implement this scheme (Bonzanni et al., 2013; Fauré et al., 2006; Saadatpour et al., 2010; Thomas, 1991). For instance, one can follow a *random order asynchronous* update rule wherein all the nodes are updated exactly once but in a random order in each iteration (also called time step). This can be done by generating a random permutation of $\{1, 2, \ldots n\}$, where $n$ is the number of nodes, at the beginning of each iteration. Alternatively, one can follow a *general asynchronous* update rule

in which the element that is updated is randomly drawn from the sequence $\{1, 2, \ldots, n\}$. Thus, some nodes can get updated, by pure chance, twice or more before another node gets its turn. These two update methods will result in outcomes that are stochastic. This is in contrast to the *deterministic asynchronous* method in which nodes are updated using a fixed sequence (Aracena et al., 2009; Mortveit & Reidys, 2007) or at pre-determined time steps set by the rates of the corresponding reaction.

For biological applications, the synchronous update scheme is most likely not appropriate; it is rare that all components in a network change their value at the same time and that all processes take the same duration of time to be completed. Asynchronous updating can in principle implement data on timing and kinetics. However, these type of data are not always available, in which case it is unclear which type of asynchronous updating rule should be used. For a comparison between synchronous and asynchronous update schemes and its consequences, we refer to a study by Fauré et al. (2006). This study applied both schemes to a model for the mammalian cell cycle and also proposed a hybrid scheme, combining both synchronous and asynchronous updating.

A third method of updating a Boolean model, also resulting in stochasticity, is through the use of so-called probabilistic Boolean networks (Shmulevich et al., 2002a). In this updating method, each node in the network has a set of update equations to choose from. At the beginning of a time step, an equation for each node is randomly chosen, after which the nodes are updated synchronously. It thus combines a rule-based determinism for Boolean networks with stochasticity arising from the uncertainty from the choice of the update equation. For a review of this type of Boolean model, including its applications, we refer to Pal et al. (2005) and Trairatphisan et al. (2013).

### 2.3. Translating rate equation models into Boolean models

To see how a signalling network may be encoded using Boolean logic, let us examine one of the simplest three-component systems

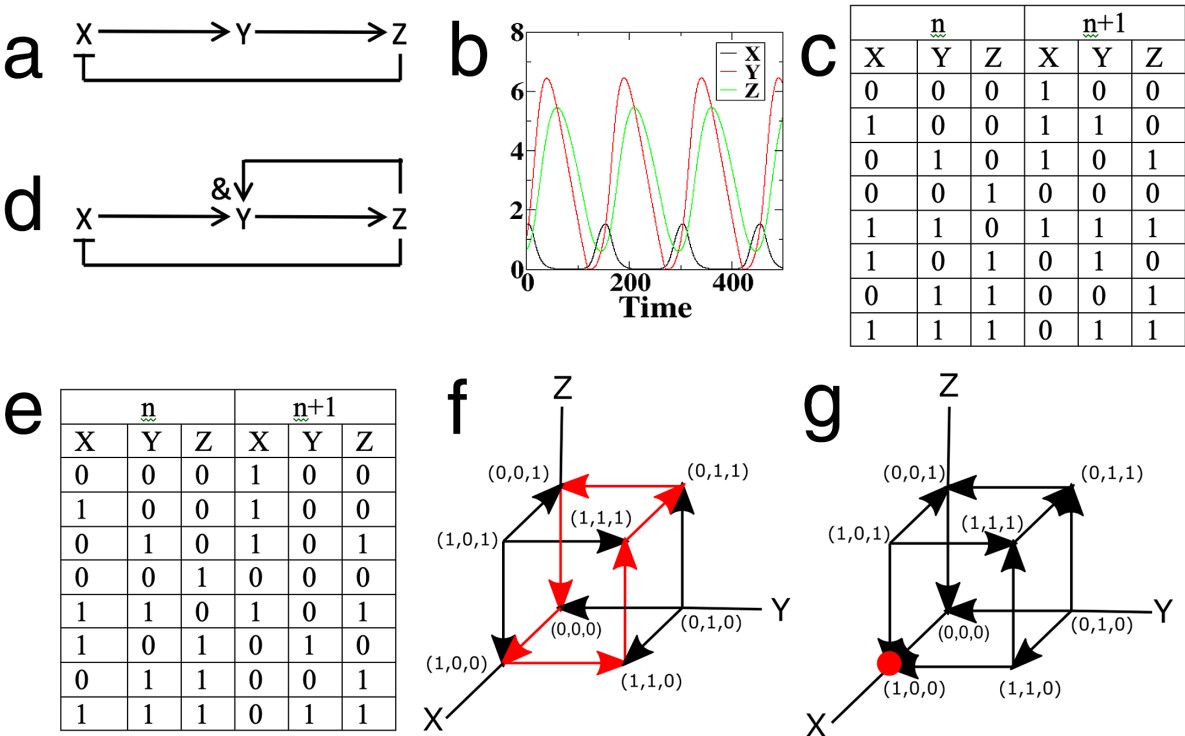

**Fig. 2.** Examples of Boolean networks. (a) Example of an oscillatory network. Arrows indicate activation and flat-edge symbols indicated inhibition. (b) The components of the network in panel (a) as a function of time, modelled using rate equations [parameters taken from Novák & Tyson (2008): $k_1$=0.1, $k_2$=0.2, $k_3$=0.1, $k_4$=0.05, $k_{-1}$=0.1, $S$=2, $K_m$=0.01, $p$=4]. (c) Truth table for synchronous updating of the network shown in panel(a) (d) Modified network in which $Y$ depends on $X$ and $Z$. (e) Truth tables for synchronous updating of the network shown in panel (d). (f,g) State space and dynamics, represented by arrows, for asynchronous updating of the networks shown in panels (a,c). Fixed point attractors are indicated by red dots while the oscillatory cycle is shown by the red arrows.

that can give rise to oscillations (Novák & Tyson, 2008). This network is shown in Figure 2a and has only three components $X$, $Y$ and $Z$. The network is wired such that $X$ activates $Y$, $Y$ activates $Z$ and $Z$ inhibits $X$. This is shown in the figures, where activation is indicated by arrows ($\rightarrow$) and inhibition by a line and a perpendicular bar ($\dashv$). This system can be translated into mathematical equations in which the concentration of the components can take on arbitrary positive values. The resulting set of ordinary differential equations is written as

$$\frac{dX}{dt} = k_1 S/(1 + Z^p) - k_{-1}X,$$
$$\frac{dY}{dt} = k_2 X - k_3 Y/(K_m + Y),$$
$$\frac{dZ}{dt} = k_4(Y - Z). \tag{3}$$

In these equations, $k_1,\ldots,k_4$ and $k_{-1}$ are the activation and degradation rates, respectively, $K_m$ is a dissociation constant, $p$ is an integer representing the non-linear inhibition of $X$ and $S$ is an input signal (Novák & Tyson, 2008). Simulating these equations for particular sets of parameters results in an oscillatory state as shown in Figure 2b.

To write this network in terms of Boolean operators, it is simplest to examine the diagram of Figure 2a. Note, however, that there are also more systematic ways to derive Boolean networks from ordinary differential equations (Davidich & Bornholdt, 2008; Stötzel et al., 2015). This diagram can be translated into the following set of Boolean operators: $Y^* = X$, $Z^* = Y$ and $X^* = $ NOT $Z$. We can then perform simulations of this Boolean network using synchronous update rules. As mentioned in Section 2.1, for

synchronous updating of small networks, it is most convenient to construct the truth table. The table for this diagram is displayed in Figure 2c, which shows that it also exhibits oscillatory cycles. Specifically, starting at $(X,Y,Z) = (0,0,0)$, the sequence is $(0,0,0) \rightarrow (1,0,0) \rightarrow (1,1,0) \rightarrow (1,1,1) \rightarrow (0,1,1) \rightarrow (0,0,1) \rightarrow (0,0,0)$, while $(1,0,1) \rightarrow (0,1,0) \rightarrow (1,0,1)$ is also a cycle.

As a second example, let us consider the previous signalling network but now changed such that the activation of $Y$ depends on both $X$ and $Z$. This can be easily incorporated by changing the rate equation for $Y$ into

$$\frac{dY}{dt} = k_2 XZ - k_3 Y/(K_m + Y) \tag{4}$$

while keeping the equations for $X$ and $Z$ unchanged. Now, there are two possible solutions: the oscillatory state, similar to the one shown in Figure 2b, and a stationary state given by $Y = Z = 0$ and $X = k_1 S/k_{-1}$. The latter is stable and the resulting state of the system depends on the initial conditions.

The Boolean network corresponding to this slightly altered network is shown in Figure 2d. The only difference between this and the previous network is that the Boolean update equation for $Y$ is now written as $Y^* = X$ AND $Z$. The truth table for this network, corresponding to the synchronous update scheme, is given in Figure 2e. This table reveals that $(1,0,0)$ is a fixed point of the system: once in this state, the network will remain in it indefinitely. Note, however, that this fixed point is only reached for certain initial conditions [$(0,0,0)$, $(1,0,0)$, $(0,0,1)$, $(0,1,1)$, and $(1,1,1)$ to be precise]. Thus, as in the continuous version of the network, the binary Boolean network displays a steady-state solution in which both $Y$ and $Z$ are zero and in which $X$ has a non-zero value.

Obviously, for the continuous system, this value depends on the model parameters while for the parameterless Boolean network it is simply one. As in the continuous system, the Boolean network also exhibits an oscillatory state: $(0,1,0) \rightarrow (1,0,1) \rightarrow (0,1,0)$, which is reached from initial conditions $(0,1,0)$, $(1,0,1)$ and $(1,1,0)$.

Let us now examine these two Boolean networks using asynchronous update rules. In this case, each element can be changed independently. Since our networks contain only three elements, this process can be visualised using the cubes shown in Figure 2f,g, where each node represents a particular state of the system and the edges represent transitions between the states. Here, the arrows indicate the transition between the different nodes according to the rules of the Boolean network. The dynamics of the Boolean network can then be determined by following these arrows.

For the network of Figure 2a, it is easy to see that the asynchronous update scheme also results in the same oscillatory cycle as the synchronous update scheme. This cycle is shown in Figure 2f by the red arrows. Contrary to the synchronous update scheme, however, the asynchronous update scheme for the second network does not exhibit an oscillatory state. For this update scheme, regardless of the initial conditions, the network always transitions to the same node (1,0,0). Thus, the steady state of the system corresponds to a fixed point, indicated by the red dot in Figure 2g. Finally, we should also point out that is possible to go "backwards" and transform a Boolean model into a continuous model (Wittmann et al., 2009). The resulting ordinary differential model could then be used to provide quantitative information regarding, for example, the concentrations of network components.

### 2.4. Encoding a Boolean network from experiments

The task of encoding a Boolean network based on experimental data is not trivial. It requires the identification of the relevant components (nodes in the network) as well as the correct update rules and thus requires biochemical, genetic and pharmacological data. While identifying components is typically not that difficult, determining the interactions between these components is challenging since the number of possible update equations grows exponentially in the number of nodes in the network (Demongeot et al., 2008). Furthermore, to define these interactions requires careful consideration of experimental data. This task is especially difficult since available experimental information is generally incomplete. To elaborate, consider a node in a Boolean network with $n$ nodes upstream. To formulate the update equation unambiguously, we need the response of the node for the whole set of $2^n$ inputs. When such information is available, formulating the equation is straightforward (Karanam et al., 2021). Generally, however, such

extensive data are unavailable and simplifying assumptions about the nature of interactions are required.

A classical algorithm to infer a Boolean network, called REVerse Engineering ALgorithm (REVEAL)(Liang et al., 1998), computes quantities encountered in information theory (Cover, 1999), such as joint entropy and mutual information. The advantage of REVEAL over earlier methods is that one only needs a small fraction of all possible input–output relations to obtain a Boolean network with a very small error rate. The method is, of course, exact when one uses all the $2^n$ input–output relations for a network of $n$ nodes. To include a more realistic scenario in which one allows for noise in gene regulation, either inherent or caused by measurement techniques, the so-called *Best-Fit Extension method* (Lähdesmäki et al., 2003; Shmulevich et al., 2001) can be employed.

We highlight here another approach used in constructing a large network, following an extensive literature search, to model guard cell dynamics in *Arabidopsis* in response to ABA (Albert et al., 2017; Li et al., 2006). Mathematically, this approach relies on developing a graph with the smallest number of nodes and edges consistent with all established qualitative relationships (Aho et al., 1972). It formulates a number of inference-based rules, shown schematically in Figure 3. In the first graph, experimental data have identified that component $A$ promotes $B$ (and is not a direct biochemical reaction) but also that $C$ promotes the interaction between $A$ and $B$. In that case, it is assumed that there is an intermediary node ($IN$) of the $A - B$ pathway and that $C$ acts on this intermediary node. If it is also known that $A$ promotes $C$, then this intermediary node can be identified as $C$ (graph 2 in Figure 3). Finally, if $A$ inhibits $B$ and $C$ inhibits the interaction between $A$ and $B$, then the logical rule can be interpreted as $A$ promotes an intermediary node $IN$, which inhibits $B$, while $C$ inhibits $IN$ (graph 3 in Figure 3). Using these rules, it was shown that the developed network was able to capture existing experimental data (Albert et al., 2017; Li et al., 2006). We will come back to this network in Section 4.1.

### 2.5. Dynamics of Boolean networks

Often, the goal of modelling is to determine the steady state of the system. That is to say, what is the outcome of the system for long times? Any deterministic Boolean model, when simulated for long enough time, converges to a limit cycle or an attractor. A limit cycle is a subset of the states of the network over which the state of the system repeats over and over in a cyclical fashion. The length of the limit cycle is the number of states in the limit cycle. An attractor is a state of the system whose 'future' state is identical to the current state; the system gets locked-in once it reaches an attractor state. We have already seen examples of these two possible outcomes when

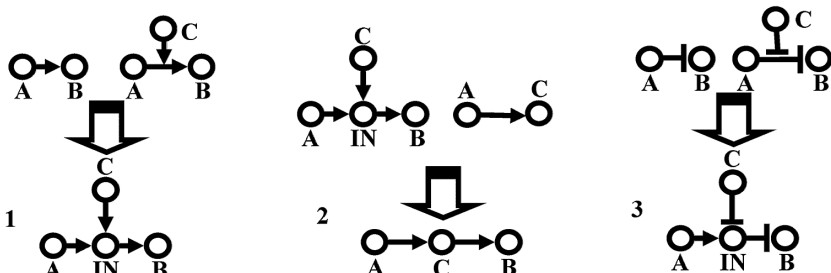

**Fig. 3.** Inference rules for the construction of Boolean networks. Experimental data are synthesised to be represented in graphs with the least number of nodes and edges, that is, as a sparse representation. This sometimes requires an introduction of an intermediary node, as in graphs 1 and 3, but when additional information becomes available, the graph can in fact simplify, as in going from graph 1 to graph 2. For further details, see text (from Li et al., 2006).

discussing the networks presented in Figure 2. The set of states that converges to a particular attractor constitutes its so-called basin of attraction. Since the evolution of the network is deterministic, no two basins of attraction share a common element; they are said to be disjoint. Likewise, a limit cycle—along with transient states that feed into it—is disjoint with the next one. Thus, the entire state space can be carved up into disjoint basins of attraction and basins of limit cycles and each trajectory of the system is a subset of the basin its initial state belongs to. Determining the number of attractors, together with their basins of attraction, is an active area of research in the mathematical field of graph theory and we direct the interested reader to several recent studies (Aracena et al., 2017; Krawitz & Shmulevich, 2007; Mori & Mochizuki, 2017; Veliz-Cuba & Laubenbacher, 2012).

Defining and analysing the basins of attraction for non-deterministic Boolean networks (e.g., using random order asynchronous or general asynchronous update methods) is not as straightforward since the trajectory of the system is no longer deterministic. Furthermore, the set of attractors and limit cycles found using asynchronous updating can be different from the ones found using synchronous updating. This was highlighted by Saadatpour et al. (2010), which carried out a comparative study of the dynamics and steady states of the ABA-induced stomatal closure network under synchronous and the three aforementioned asynchronous update schemes. To enumerate the fixed points and limit cycles, they reduced the system using Markov chains (Ross, 2014) and by simplifying the Boolean update equations. They found that both types of update schemes exhibited a fixed point. However, for synchronous updating, they found large basins of attractions for two limit cycles. These limit cycles, and their basins of attractions, were not found using asynchronous updating, unless strict limitations regarding the timing of several processes were implemented.

### 2.6. Software tools

Once a network and the update schemes are defined, a Boolean network can be simulated to obtain the trajectory and steady states of the system, to visualise the network, and to determine the activity levels of its components. A large number of computational tools have been developed to simulate Boolean networks on personal computers, as reviewed recently by Schwab et al. (2020). In addition, software packages are available to determine and computationally identify the attractors of a Boolean network (Rozum et al., 2021; Rozum et al., 2022). Most of these packages, but not all (Klamt et al., 2007), are open source and can thus be freely used. Some of these tools, however, do not use a graphical interface, which makes it more challenging to construct and visualise the network (Albert et al., 2008; Garg et al., 2008; Helikar & Rogers, 2009; Klarner et al., 2017; Müssel et al., 2010; Paulevé, 2017; Stoll et al., 2012). Other packages only allow synchronous updating and can thus not implement an asynchronous update scheme (Bock et al., 2014; Terfve et al., 2012). Finally, some packages are only able to run a single initialisation at a time, which means that probing a large set of initial conditions, especially valuable for large scale networks with asynchronous updating, is challenging (Gonzalez et al., 2006; Schwab & Kestler, 2018).

We have recently developed Boolink, a simulation platform for Boolean networks that is based on a graphical user interface and is completely open-source (Karanam et al., 2021). Specifically, the software allows users to define the nodes and connections in the Boolean network, visualise the network as a tree, set various

simulation parameters including the number of time steps and initial conditions, plot the activity of a few chosen nodes, and to analyse the trajectory of the system as a whole. Boolink is written in Python and C++, and the source code is freely available from the GitHub repository, https://github.com/rappel-lab/boolink-gui, along with its documentation, to use, modify and distribute. We have also packaged the software as a Docker container (Merkel, 2014), which is a self-contained system that comes with all the software dependencies and runs straight out of the box. In its original presentation, Boolink was only able to simulate a Boolean network using the physiologically relevant asynchronous update scheme. Recently, however, we have extended Boolink to include the ability to simulate networks using a synchronous update scheme.

## 3. Boolean networks and gene regulation in plants

Creating a regulatory framework based on the available data on gene expression is essential to understanding gene expression. A network that is inferred from gene expression data is termed GRN (Emmert-Streib et al., 2014). Several methods have been developed to construct GRNs from available data, including Boolean models, information-theory-based models, and machine-learning-based models. These methods and their application and suitability in different contexts is an active field of research and we refer the interested reader to several review articles (Chai et al., 2014; Delgado & Gómez-Vela, 2019; Fiers et al., 2018; Zhao et al., 2021). Here, we limit our discussion to Boolean models.

The first application of Boolean modelling was carried out by Kauffman when he described a genetic network (Kauffman, 1969a). In a Boolean gene network, a gene is either turned ON (i.e., has value 1) or turned OFF (with value 0), while the topology of the network specifies how and if a gene interacts with other genes. In plants, Boolean networks have been applied to a number of genetic networks. We will discuss here three different examples: Boolean models (a) for flower development, (b) for induced systemic resistance (ISR) induced by microbes and (c) for the root stem cell niche (SCN). These models have introduced modifications to the simple implementations of Boolean networks described so far. These modifications will be discussed as the systems are introduced.

### 3.1. Flower development

One of the first examples of Boolean modelling studied early flower development in the model plant *Arabidopsis thaliana* (Mendoza & Alvarez-Buylla, 1998). In this model, 12 genes were considered and the topology of the network was determined based on experimental data. The model was slightly more involved than the simple Boolean implementation we described in Section 2 in that the modified model is known as a threshold Boolean model. Each node of the model still takes binary values (0 or 1) but interactions between any pair of nodes are encoded by *weights* between them; excitatory interactions carry a weight of +1 whereas inhibitory interactions carry a weight of −1. The update equation of a node in this model is not Boolean but algebraic, consisting of the sum of weighted interaction terms. When a node is updated, the sum of all of its interactions with other nodes is calculated. If the sum exceeds the *threshold* of the node, then the node is updated to 1; if not, to 0.

The update scheme for this model is in-between the synchronous and asynchronous update schemes as described before, and is termed semi-synchronic, and block-sequential and block-parallel in later iterations (Aracena et al., 2009; Demongeot & Sené, 2020). Instead of updating all the nodes at once or one after another

in some order, the nodes in the model are grouped into *blocks*. All the nodes in a block are updated at once, and the blocks themselves are updated sequentially. This method makes use of qualitative experimental data, such as the order of activation of different parts of the genetic network.

The model was found to have six attractors, four of which were consistent with the gene expression patterns observed in *A. thaliana* (Mendoza & Alvarez-Buylla, 1998). One of the remaining two was not able to flower and the sixth one, while not observed, could be induced experimentally (Mendoza & Alvarez-Buylla, 1998). Since this Boolean threshold model was published, several studies have further analysed its dynamics. These studies revealed that it is possible to reduce its complexity while maintaining its steady-state behaviour (Demongeot et al., 2010; Ruz et al., 2018) and highlighted the crucial role of the plant hormone gibberellin in normal flower development (Demongeot et al., 2010).

## 3.2. Induced systemic resistance

Recently, the ISR in *A. thaliana* plants triggered by beneficial microbes was investigated using Boolean modelling (Timmermann et al., 2020). ISR is an important defense mechanism of plants against harmful pathogens (Pieterse et al., 2014) and the study investigated how the bacterium *Paraburkholderia phytofirmans* PsJN can trigger ISR and protection from the bacterial pathogen *Pseudomonas syringae* DC3000 (Timmermann et al., 2017, 2019). It used the temporal experimental expression patterns of eight key genes following inoculation of PsJN and asked which threshold Boolean network was able to reproduce the time series data. Parameters of the model, including the weights among the nodes and their threshold values, were fitted to experimental data using an algorithm called differential evolution (Storn & Price, 1997), which belongs to a class of fitting algorithms called genetic algorithms (Ruz et al., 2015). The study inferred 1,000 networks from the data. One of these networks was chosen and pruned using biological reasoning. The robustness of the pruned network was then tested by determining how mutations of fundamental genes affected the ISR response. These virtual mutation experiments produced responses that were consistent with available experimental data (Timmermann et al., 2020). Additionally, the study found that the pruned consensus network is robust because it requires an unlikely event of a triple mutation

to the network before the ISR is lost. Furthermore, the authors argue that, in the presence of errors in gene expression data, the differential evolution algorithm used to derive the GRN fared better than classical algorithms to infer Boolean networks, including REVEAL (Liang et al., 1998) and best fit extension (Lähdesmäki et al., 2003).

## 3.3. Root stem cell niche

The examples above applied Boolean modelling to determine the most probable network that is consistent with experimental data. In doing so, these studies found missing links or were able to determine the most critical network components. As a result, these Boolean models were often able to predict novel components or connections between components and could suggest new experiments. To further illustrate the ability of Boolean models to guide experiments, we focus here on another example of a gene network studied using Boolean modelling, the root SCN in *A. thaliana* (Azpeitia et al., 2010; Velderraín et al., 2017). The root SCN in *A. thaliana* is well studied and is located at the root apical meristem (Dolan et al., 1993). It consists of a so-called quiescent center (QC), comprised of four infrequently dividing cells, and, in immediate proximity, active stem cells that are called initials. The divisions of these initials result in different types of differentiated cells and in tissue growth of the plant (Dolan et al., 1993). The question thus arises, how can the undifferentiated cells of the QC give rise to several differentiated cell types?

Modelling, and in particular Boolean modelling, is ideal to address this question. Experimental work has identified a number of molecular and genetic components that play a role in the maintenance of the SCN (Pardal & Heidstra, 2021). Furthermore, the interactions between some, but not all of the components can also be deduced from experimental work. It is therefore possible to construct a putative wiring diagram as in Figure 4a, which shows the components along with their interactions as either arrows, indicating activation, or flat-end symbols, indicating repression. The dynamics of this network should then allow steady-state solutions with gene expression that is consistent with the different cell types of the SCN. In terms of Boolean modelling, this means that the network should display attractors corresponding to these different cell types.

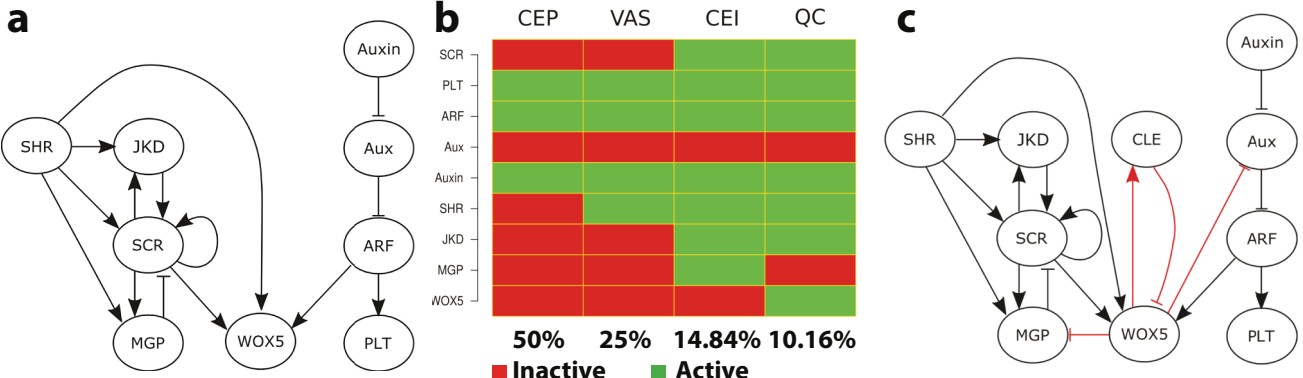

**Fig. 4.** Boolean modelling of gene networks. (a) Example of a putative network that maintains the SCN in *Arabidopsis*. Arrows indicated activation and flat-edge symbols correspond to repression. For the definition of the different components, see Velderraín et al. (2017). (b) Attractors of the Boolean network shown in panel (a). Green represents an active and red represents an inactive gene. The labels at the top of the diagram represent the attractors and correspond to the phenotypes observed in experiments (CEI, cortex-endodermis initials; CEP, columella epidermis initials; QC, quiescent center; VAS, vascular initials) (from Velderraín et al., 2017). (c) Modified network based on novel experimental and computational results (from Velderraín et al., 2017).

Simulating the Boolean network in Figure 4a using synchronous updating revealed four different attractors as shown in Figure 4b. In this diagram, active genes are displayed in green while inactive genes are displayed in red. Each of the four attractors correspond to a different set of genes that are ON or OFF and, thus, to a different cell phenotype. For example, the gene SCR (SCARECROW) is ON (and thus has a value of 1) in the phenotype corresponding to columella epidermis initial and vascular initial but is OFF and has value 0 in cortex-endodermis initial and QC (QC cell). The diagram also shows the size of the basin of attraction, expressed as the percentage of initial conditions that resulted in the attractor.

Further testing of this model and comparing the outcomes to experimental results showed that certain interactions were missing. For example, this analysis revealed the need for a repressor of WOX5 (WUSCHEL RELATED HOMEBOX 5) and an additional component with an inhibitory link was predicted (Azpeitia et al., 2010). This prediction was verified in experiments, which showed that WOX5 is negatively regulated by CLE40 (CLAVATA-like-40) (Stahl et al., 2009). Additional predictions resulted in the modified network displayed in Figure 4c, where the postulated interactions are shown in red (Azpeitia et al., 2010). After this study and once new experimental findings became available, this network has been modified and extended further (Azpeitia et al., 2013). These studies showed the power of Boolean modelling: once a Boolean network has been constructed, it is fairly straightforward to modify and extend it and to generate experimental predictions. These modifications and extensions are much easier to implement than in continuous models based on rate equations. In those type of models, a modification typically requires refitting and adjusting the model parameters, which can be an arduous task (Karmakar et al., 2021).

## 4. Boolean networks and signalling in plants

Biological signalling pathways can be very complex, containing numerous components and multiple feedback loops. Such complex pathways can also be addressed by Boolean modelling and examples include T-cell signalling (Saez-Rodriguez et al., 2007), molecular pathways of neurotransmitters (Gupta et al., 2007) and cancer pathways (Fumia & Martins, 2013; Sherekar & Viswanathan, 2021). A prime example of a complex signalling network is found in plants, where the network regulating phytohormone ABA-induced stomatal closure contains a large number of interconnected components. Below, we will review studies that attempt to cast this closure pathway into a Boolean network. Furthermore, we will also discuss recent efforts to extend this signalling network to include $CO_2$ signalling.

### 4.1. ABA signalling network

Stomata are pores in the epidermis of leaves that regulate gas exchange, including $CO_2$ for photosynthesis and loss of water vapor. Each stomata is formed by a pair of guard cells and its aperture is modulated in response to environmental changes such as light and $CO_2$ (Assmann & Jegla, 2016; Munemasa et al., 2015). Furthermore, drought results in the accumulation of ABA in guard cells, which leads to stomatal closure (Hsu et al., 2021; Raghavendra et al., 2010).

The network that underlies ABA-induced stomatal closure in *A. thaliana* is complex and contains a large number of components (>80). Consequently, the number of rate constants is also very large and, not surprisingly, many are not quantified. To illustrate the complexity of the network, we reproduce in Figure 5, the network

investigated by Albert et al. in a recent study (Albert et al., 2017). Given the complexity and the number of components, this network is particularly suitable for Boolean approaches (Albert et al., 2017; Li et al., 2006; Maheshwari et al., 2019; Maheshwari et al., 2020; Waidyarathne & Samarasinghe, 2018).

Albert and colleagues encoded the ABA-induced stomatal closure pathway into a Boolean network. This network has a single input node, representing ABA, and a single output node, representing stomatal closure. Obviously, both the input and output node were also taken to be binary: ABA is either 1 (present) or 0 (absent) while a similar logic applied to the closure node. The network shown in Figure 5 was constructed following a careful review of available experimental literature. The Boolean dynamics was simulated using an asynchronous update scheme in which nodes were updated in a randomly selected order at each time step (random order asynchronous). A large number of simulations (2,500) were performed and the percentage of closure was computed as the percentage of simulations in which the node Closure=1 in the steady state. A verification of the wild-type model, that is, without any alterations to the wiring or components, was performed by computing the percentage of closure with and without ABA present. The results are shown in Figure 6, as expected, in the absence of ABA, the stomata remained open and the percentage of closure was 0% (open circles) while in the presence of ABA the stomata close (closed circle; 100% closure).

One of the strengths of Boolean modelling is the ease with which one can alter the network and either 'knock-out' a node or make a node 'constitutively active'. A virtual knock-out experiment equates to fixing one of the nodes to 0 while fixing a node's value to 1 corresponds to making this component constitutively active. Albert et al. performed a systematic analysis of the response of the Boolean network to ABA exposure in which all internal nodes of the network were set to either 0 or 1. Some of the results of these knock-out experiments are shown in Figure 6. For example, knocking out cytosolic pH resulted in reduced sensitivity to ABA with only 35% closure, consistent with experiments (Zhang et al., 2009).

The study showed that altering single nodes in the networks resulted in either an increased, a decreased, or an unchanged sensitivity to ABA. Where possible, the results were compared to available experimental data and this comparison agreed in most cases. In the case where no experimental data were available, the computational result could be considered a prediction. Some of these predictions were subsequently tested using experiments, demonstrating the usefulness of Boolean modelling.

Even though the model was able to reproduce experimental data in more than 75% of the predictions, there were several clear discrepancies with experimental observations (Albert et al., 2017). Further highlighting the strength of Boolean modelling, these discrepancies were used in a follow-up study, which aimed to improve the model (Maheshwari et al., 2019). For this, the original model was first reduced to a smaller network with 49 nodes and 113 edges. This reduced network was shown to duplicate all results from the original network. A subsequent computational analysis of this reduced network then revealed that that inhibiting PP2C protein phosphatase ABSCISIC ACID INSENSITIVE 2 (ABI2) by cytosolic calcium was able to rectify most of the discrepancies. The proposed inhibition by calcium was also verified in experiments, highlighting the ability to iterate between model and experiment (Maheshwari et al., 2019). This iterative quality of Boolean modelling was also evident from a recent and additional follow-up study (Maheshwari et al., 2020). This study examined the response of the improved

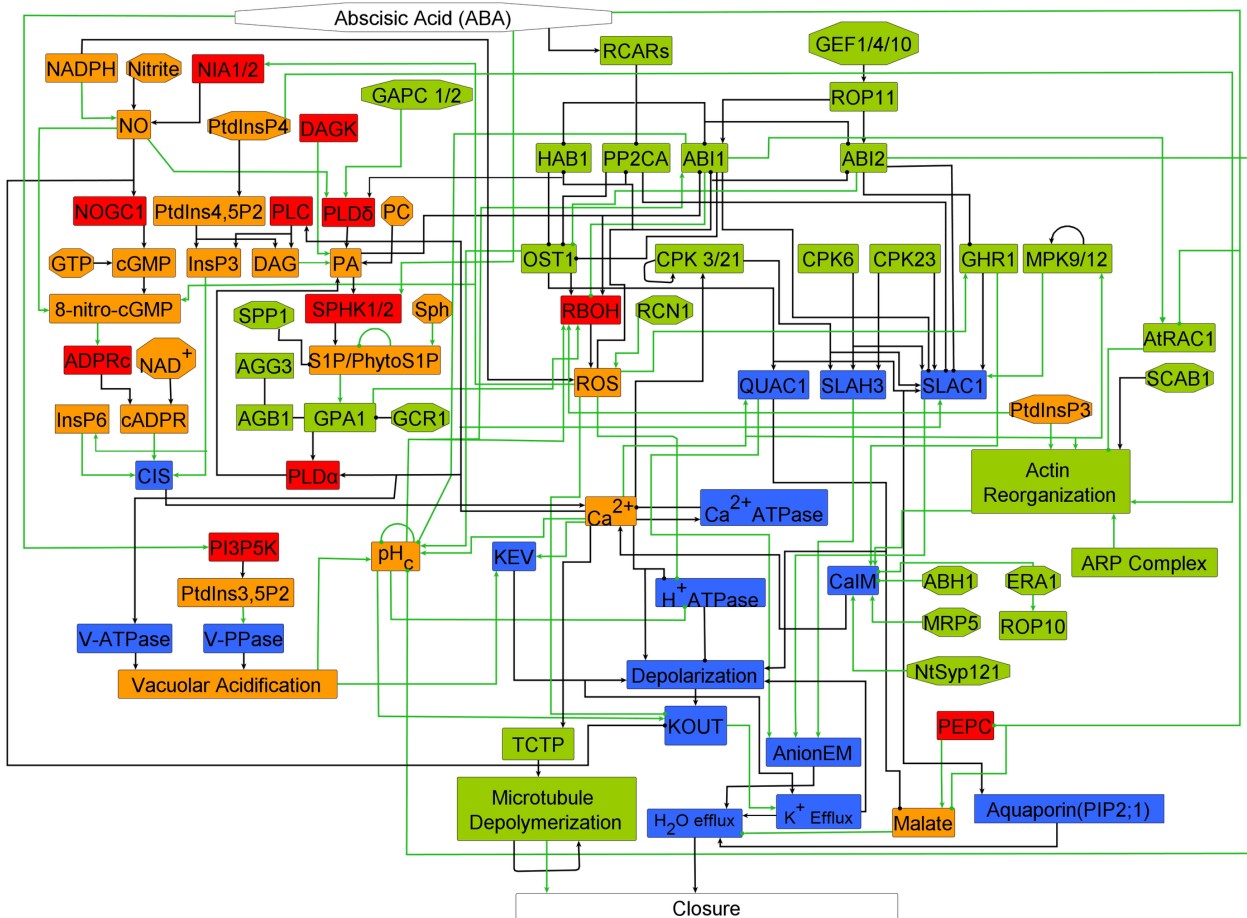

**Fig. 5.** Signalling network of abscisic acid (ABA)-induced stomatal closure. Arrows indicate positive interactions while filled circles indicate negative interactions. Rectangles represent nodes that are connected to other nodes. Black lines represent direct interactions and green lines represent indirect interactions. Nodes are color coded according to their function: enzymes (red), signalling proteins (green), membrane-transport related nodes (blue), and secondary messengers and small molecules (orange). For names of the components, see original publication (Albert et al., 2017).

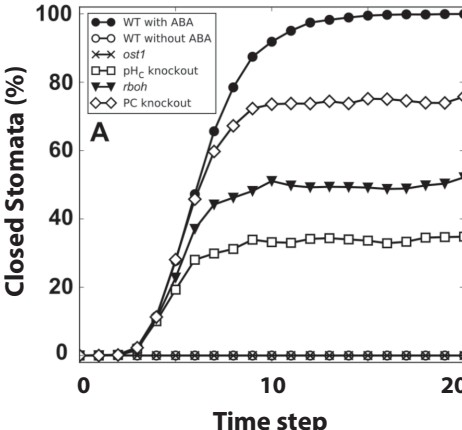

**Fig. 6.** Results of a Boolean ABA network. Shown are the percentage of closure as a function of iteration step. The wild-type (WT) curves show the network response in the absence of (open circles) and presence of ABA (closed circles). Other curves show the response following simulated knockout of the component (node set to 0) in the presence of ABA. For abbreviations, see original study (Albert et al., 2017).

network in the absence of ABA or following the removal of ABA. In the first case, the stomata should remain open while in the second case, their closed state should relax to an open state. Probing the

network under these conditions, it was further improved so that its response was consistent with experiments (Maheshwari et al., 2020).

### 4.2. CO$_2$ signalling pathway

The Boolean network shown in Figure 5 was recently extended to include CO$_2$ regulation of stomatal movements (Karanam et al., 2021). This extension is possible since several elements of CO$_2$ signalling overlap with those of ABA signalling (Hsu et al., 2018; Merilo et al., 2015; Zhang et al., 2020). Elevated CO$_2$ levels result in the closure of stomatal pores and thus affect the water use efficiency and yield of crop plants (Dubeaux et al., 2021; Engineer et al., 2016; Zhang et al., 2018). In an initial attempt, the ABA network was complemented with a CO$_2$ branch as shown in Figure 7a. This branch consisted of CO$_2$ as an input node and, through several intermediary nodes, inhibited the GHR1 (guard cell hydrogen peroxide resistant 1) node in the ABA network. As in the ABA studies, Boolean dynamics was implemented using asynchronous updating using randomly selected nodes (random order asynchronous) and results were averaged over a large number of realisations (Karanam et al., 2021). The response of ABA under high and low CO$_2$ conditions, modelled by setting the node CO$_2$ to either 1 or 0, is shown in Figure 7b. Note that here the conductance level is used instead

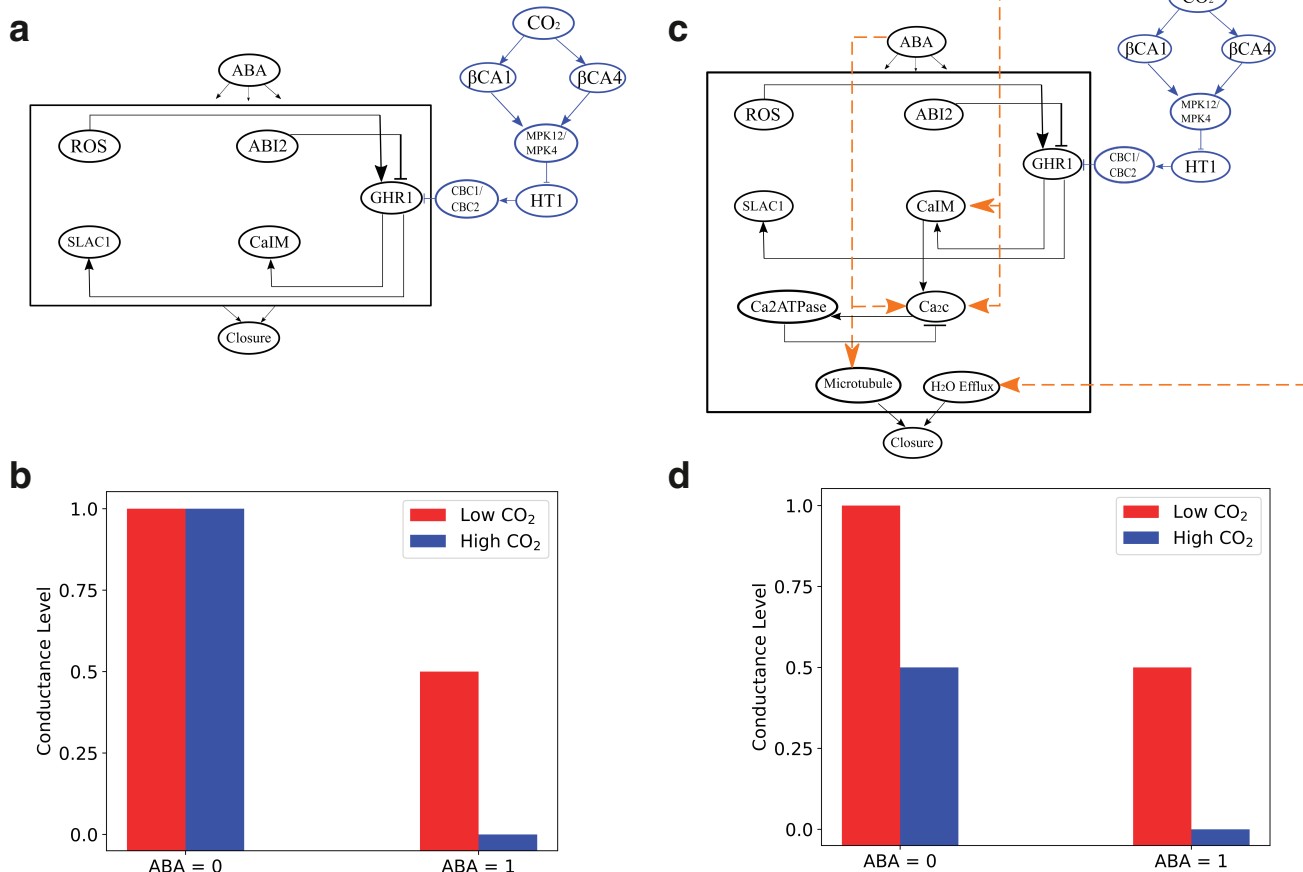

**Fig. 7.** $CO_2$ and ABA-induced stomatal closure model. (a) Extended network showing the new $CO_2$ branch in blue and the existing ABA network, shown in Figure 5, as a box. Only several of ABA components are shown. (b) Predicted stomatal conductance levels of the network in panel (a) for both $CO_2=0$ (red) and $CO_2=1$ (blue), before and after the application of ABA. (c) Modified $CO_2$ and ABA-induced stomatal closure network, with modifications represented by orange links. (d) Predicted stomatal conductance levels using the network shown in panel (c) for both $CO_2=0$ (red) and $CO_2=1$ (blue), before and after the application of ABA. For names of the components, see original publication (Albert et al., 2017) (from Karanam et al., 2021)

of the percent closed stomata. This conductance level is simply computed as 1-Closure. The simulations predicted that the introduction of ABA leads to a decrease of conductance level from 1 to 0 when $CO_2=1$, corresponding to fully closed stomata. For $CO_2=0$, the exposure to ABA resulted in a reduction of the conductance level from 1 to 0.5.

These predictions were subsequently tested in experiments by determining ABA-mediated stomatal closure under high and low $CO_2$ conditions (Karanam et al., 2021). These gas-exchange experiments were conducted by applying ABA to the transpiration stream of excised intact leaves (Ceciliato et al., 2019). The experiments revealed that the application of ABA under both conditions resulted in a reduced conductance level (Karanam et al., 2021). In contrast to the simulation results of Figure 7b, however, the steady-state stomatal conductance prior to ABA application was different for high and low $CO_2$. Specifically, it was higher for low $CO_2$, demonstrating that $CO_2$ induces stomatal conductance reduction. Taken together, these experimental results suggested that both $CO_2$ and ABA reduce stomatal conductance and that they have additive responses (Karanam et al., 2021).

Again showing the ability to iterate between experiments and modelling, the network was modified to account for the experimental results. The updated network topology is shown in Figure 7c where the added links are displayed in orange. These links were

partially motivated by experimental data. For example, adding $CO_2$ dependence on calcium signalling was motivated by experiments that showed that cytosolic calcium is involved in $CO_2$-induced stomatal closure (Schulze et al., 2021; Schwartz et al., 1988; Webb et al., 1996). The response of the updated network to ABA application is shown in Figure 7d and is now consistent with experimental results. Importantly, in the absence of ABA, the updated network has a steady state that depends on the $CO_2$ level. Furthermore, application of ABA resulted in absolute conductance changes that were similar for both conditions (Karanam et al., 2021).

## 5. Conclusion and outlook

This article summarises recent attempts in modelling genetic networks and signalling pathways using Boolean models. We have focused on plants, where this type of modelling has been used in a wide variety of studies. We have attempted to outline how one can transform a model composed of ordinary differential equations into a Boolean model and have described methods to synthesise experimental data into logical equations. Furthermore, we have shown how different update rules can result into different outcomes and have pointed readers to available software for Boolean modelling. We have also reviewed a number of genetic and signalling networks in plants that have been investigated using

Boolean models. In doing so, we have strived to highlight the advantages and promises of Boolean modelling.

Most importantly, Boolean models offer a simple yet effective way to model the steady states of reaction networks in biological systems made of a large number of components. They only require knowledge about the components of the network and the nature of the connectivity between components but not detailed knowledge about kinetics or rates. Boolean models are therefore especially useful in cases where the current knowledge of interactions is only qualitative, when the kinetic and rate parameters are not precisely determined, or when the network is too large to be simulated in a reasonable time. We should point out, however, that constructing a Boolean network is not the only way to simplify a complex mathematical model. Especially when a model falls into the class of so-called sloppy models, in which many of the parameters are loosely constrained (Brown & Sethna, 2003; Gutenkunst et al., 2007), it is sometimes possible to systematically reduce the number of equations while still maintaining the predictive value of the model (Lombardo & Rappel, 2017; Transtrum & Qiu, 2014).

One of the main strengths of Boolean modelling is the ease with which one can generate experimental predictions. As we have discussed here, these predictions have resulted in the identification of critical components or interactions in both genetic networks in plants [e.g., negative regulation of WOX5 by CLE40 (Azpeitia et al., 2010)] as well as in signalling networks [e.g., calcium inhibition of ABI2 in the ABA signalling pathway (Maheshwari et al., 2019)]. These predictions can then be tested in experiments, further specifying the underlying networks and improving our understanding of the biological mechanisms. Boolean modelling, however, is not well suited to address detailed kinetics of networks and pathways. After all, time does not explicitly occur in a Boolean model and most applications therefore focus on determining steady states of the system. Furthermore, Boolean networks cannot be used to model graded outcomes such as concentration of network components that are different from 0 and 1. In order to achieve that, hybrid models have been developed in which the set of states is expanded beyond $\{0,1\}$ and the update equations contain a combination of algebraic and Boolean terms (Glass & Kauffman, 1973). These hybrid models have been used to model the interaction of immune cells and pathogens (Thakar et al., 2009), gene networks underlying flower development in *Arabidopsis* (Mendoza & Xenarios, 2006), and light-induced stomatal opening in plants (Sun et al., 2014). Despite these limitations, we expect that Boolean modelling will continue to play a prominent role in deciphering the structure and mechanisms of biological networks in general and of plant networks in particular.

**Financial support.** This research was funded in part by a grant from the National Science Foundation (MCB-1900567).

**Conflict of interest.** The authors declare no conflict of interest.

**Authorship contributions.** W.-J.R. conceived the outline of the review and coordinated the writing process. A.K. extended the Boolink software. W.-J.R. and A.K. wrote the article.

**Data availability statement.** No new data or code is presented in this article.

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
