## [Reviewer Report]

Dear Dr. Hamant, 

My co-author Aravind Rao Karanam and I are delighted to submit our review entitled “Boolean 

modeling in plant biology”. As you will recall, this submission was initiated by your kind 

invitation, which you sent last year after reading our recent work on Boolean modeling. We 

believe we have created a review that is of interest not only to the readership of your journal

in particular but also to the broader plant biology community. 

We look forward to your response.

Best regards,

Wouter-Jan Rappel

Wouter-Jan Rappel, Ph.D. 

Department of Physics 

UC San Diego

9500 Gilman Drive 

La Jolla, CA 92093

---

## [Reviewer Report]

*Comments to Author*: The authors have reviewed the use of Boolean network modelling in plant biology, giving a pedagogical introduction and a set of examples from the field. The review is well written and this will serve as a useful reference piece, and is very aligned with QPB's mission. I do have some comments about the style and structure that I think may help further improve things, outlined below. For transparency, I am Iain Johnston and am happy for this review to be treated as public domain. To my mind, my most important limitations as a reviewer here are:

- I come from the same disciplinary background as the authors, and so may not be the best person to comment on how readily this article may be read by biologists unfamiliar with these ideas (although this is my major comment)

- There are several papers in the authors' set of examples with which I'm not familiar.

Awkwardly, and related, my most important point isn't a single actionable change. At the moment the review is a very good introduction to Boolean modelling in plant biology -- *for physicists*. There are several clear instances where the target audience of the article appears to be physicists, not biologists. This is apparent in, for example, some jargon (“disjoint”, “sparsest graph”); leaning on ODEs as a pedagogical “stepping stone” to introduce an example. Some of these words -- and ideas -- will be unknown to the biologists reading the article as an introduction to the topic.

How best to address this? A glossary would be one option. Another would be to give a layman's introduction to such terms and ideas as they arise -- though this would need more words. Other options may be possible too. But perhaps a useful exercise might be to read through the article from the perspective of a biology PhD student. They've done maths maybe up to the age of 18, they've heard of ODEs but don't gain immediate intuition about a system from looking at them, they haven't done graph or set theory, and are more used generally to thinking qualitatively than quantitatively. Can their hands be held more through the concepts that are bread-and-butter to physicists, but unusual to biologists?

Other points --

The authors focus on logic gates as the key architectures in BNs, with expressions like C = A AND B. A couple of thoughts:

1. The equality sign is a bit odd here (and throughout), because we're fundamentally talking about update rules. Perhaps C -> A AND B, or C_new = A AND B would make this clearer? (as before the update rule is applied, the equality doesn't necessarily hold)

2. Can the authors include the (albeit rather trivial) identity operation (A -> B) in their introduction? This is extremely common in biology (transcription factors acting as activators) -- and indeed comes up in their examples.

The introduction doesn't mention parametric Boolean models like Boolean Threshold Dynamics, where edges are given (positive or negative) weights which act as coefficients for the incoming node states in a function that determines the update step. Although any static instance of such a model can (of course) be mapped to a model using logic gates alone, the parametric versions are commonly used and merit an explanation.

Fig 3 is pretty unclear as a standalone object. Can the experimental and inferred structures be distinguished (eg by colour) and the caption expanded? “For details, see text” is awkward for a reader viewing the figures as an extended abstract. Also -- are the figures taken directly from the source papers? There may be copyright issues here if so? Some other figures -- Fig 7 in particular -- aren't as good as they could be. In Fig 7 B,D the text is tiny, lots of whitespace, and the takehome from the time series isn't immediately apparent from the figure.

“Boolean networks and plant genetics” is an odd title, and the reference to “genetics” throughout also sit awkwardly. Most readers will think inheritance, chromosomes, breeding, mutations when reading “plant genetics”. I suspect the emphasis the authors mean is “gene *regulatory* networks”. If so, can “genetics” be replaced throughout with “gene regulation” or similar?

The point on p13-14 about the computational ease of knockouts or other manipulations probably deserves promotion to the introduction, as it's a key and general strength of the modelling approach.

If I may, I'd like to suggest a classic citation that would point the interested reader both to more information about Boolean modelling and the broader spectrum of modelling approaches for gene regulation: https://www.nature.com/articles/nrm2503

Did the ISR case study do anything more than match experimental data? If there was additional insight that came from the modelling approach it'd be nice to hear about it.

Some Bibtex bugs mean that several internal references aren't clear (“Sec.”).

---

## [Reviewer Report]

*Comments to Author*: Although Boolean networks are simple models of gene regulatory networks, they help capture qualitative information about the gene regulatory processes. Also, the model presents very interesting mathematical properties. In this sense, I believe the submitted review paper is of interest, contributing to this field.

For completeness purposes, I think the following sections need to improve with the following comments:

Update rules

1-It is important to point out in this section that the number of possible updating rules is exponential. Indeed, for a network with n nodes, the number of updates is given by a recursive formula (Proposition 5) that appears in:

Demongeot, J., Elena, A., Sené, S., 2008. Robustness in regulatory networks: A multidisciplinary approach. Acta Biotheoretica 56, 27–49.

For example, for n=10, there are 102,247,563 different updating rules.

2-A popular updating rule in biological applications, like the Arabidopsis thaliana network, is the Block-Sequential updating rule which is not discussed in this section. In this case, the set of nodes for a given sequence is partitioned into blocks. The nodes in the same block are updated synchronously, but blocks follow each other sequentially. For example, in the Arabidopsis thaliana network, the updating rule is (EMF1, TFL1)(LFY, AP1, CAL)(LUG, UFO, BFU)(AG, AP3, PI)(SUP).

Encoding a Boolean network from experiments

3- A review of classical methods to infer Boolean networks from data is missing, like REVEAL (and variants), Best-Fit extension algorithm, etc.

4- A review of more recent approaches to infer Boolean networks from data using evolutionary computation is missing.

Dynamics of Boolean Networks

5- Derterming or counting the number of fixed points a Boolean network can have is an active research topic. There have been several works published in this field. Authors should consider reviewing some.

---

## [Reviewer Report]

*Comments to Author*: Dear Authors

Thank-you for your submission to QPB. We have received 2 expert reviews on this manuscript providing constructive feedback on how to improve the text.

The manuscript is well written and organized, covering relevant topics on Boolean modelling towards understanding processes in plants. The comments of reviewer 1 are particularly pertinent in terms of the extent to which the target audience has background knowledge in maths and graph/set theory. An effort to simplify the language and presentation of the material would enhance the reach and penetrance of the work.

We look forward to reading this manuscript in a revised form.

---

## [Reviewer Report]

*Comments to Author*: To my eyes the manuscript is substantially more accessible from a biologist's perspective, and I believe the authors' edits have clarified several points. Happy to recommend acceptance, and thanks to the authors for a useful reference -- I will be using it!

One remaining point from me. The topic of inferring GRNs from data has now been raised. There could be (and is) a whole review article on this topic alone. I recognise that this is not the focus of this article but the methods that the authors mention in their response to Reviewer 2 seem rather sparse and dated (early 2000s) -- the field, and technology, has advanced dramatically since then. I recommend including a link to a recent review on the topic for the interested reader to follow. I notice (without endorsing it, as I haven't read it) that this reference exists for example

https://academic.oup.com/bib/article-abstract/22/5/bbab009/6128842

---

## [Reviewer Report]

*Comments to Author*: The revised version satisfactorily addresses my previous comments, good work.